# GENERATING FAKE DATA TO FAKE PRIVACY PRYERS

## ABSTRACT

Asymmetry of data complexity and model capacity can create privacy vulnerability. That is because if there are relatively fewer data points while the model capacity is relatively higher, a model may memorize almost all the data points. As a remedy for the issue, more data samples can be generated. When generating more data samples, the aim is to protect and promote the original data as privacy-safe as possible while generating more privacy-risky data samples to fake privacy attackers. To enable the aim, we investigate each individual data sample's privacy level, unlike existing studies that only take into account an overall dataset's privacy, which is not precisely effective. We show how effective our generative approach is in combating privacy attacks. Our work is novel in that we propose a sample-level valuation, and data transformation and generation approach in the privacy domain.

## 1 INTRODUCTION

Over the past decade or so, the advancement of machine learning models on various challenging datasets has mostly relied on the increase in model capacity. However, in practice, the growth in data complexity often cannot keep pace with the increasing model capacity. Once such an imbalance between data complexity and model capacity is present, a model tends to fully memorize almost all of the data points. In particular, such memorization directly puts memorized samples under privacy risks, although it yields little or no contribution to the performance of the model.

Straightforward solutions to this issue are generating more fake data samples so the model can properly "learn" most of the samples instead of memorizing them. When generating more data samples, the aim is to protect and promote the original data samples as privacy-safe as possible while generating more *fake* and privacy-risky data samples.

Regarding data privacy, existing studies only take into account an overall dataset's privacy but not an individual data sample's privacy safety levels. Unlike them, we investigate each individual data sample's privacy safety and vulnerabilities. We valuate each sample's privacy level to categorize them being privacy-safe, privacy-vague, or privacy-risky. Privacy-safe samples leak privacy information the least, while privacy-risky samples are most vulnerable to privacy attacks. Privacy-vague samples are either privacy-safe or privacy-risky, depending on the models or experimental settings.

By investigating the sample-level privacy, we observed that the privacy of a model depends on different data complexities. That is, given a fixed model capacity, changes in data complexity will change the proportions of these three types of samples: a dataset with higher complexity tends to contain a higher proportion of privacy-safe samples. In particular, our data valuation approach discovered that there are samples tending to be privacy-safe more preferentially than others across various independent training dynamics, indicating there are certain features that push the model to memorize the specific sample. With those insights in mind, we propose a generative approach for privacy domain transformation. It transforms samples into desired privacy domains so that the model can or cannot memorize samples with proper priorities. Through such transformations, our method can generate more fake privacy-risky data to occupy the model's memory capacity and keep the original privacy-safe data from being overly memorized.

Here is a brief overview of our novel observations and contributions:
**Privacy vulnerability can arise due to asymmetry of data complexity and model capacity.** We identified privacy vulnerabilities that arise when a too-high-capacity model memorizes almost the entire dataset which is with relatively too few data points.

**A higher sample count does not solely help privacy, but more distinctive samples are required for privacy protection.** This is why mere augmentation alone cannot help privacy. And that's why we do generate and transform samples to generate more independently different samples.

**The more privacy-risky fake samples can more confuse attackers and prevent the model from memorizing.** We show that generating more privacy-risky (fake) samples in the trainset makes the task more challenging for the attacker. The extensive empirical results prove this insight.

## 2 RELATED WORK

**Privacy Defense on Machine Learning**    Nasr et al. (2018) proposed an adversarial training framework to defend against membership inference attacks by aligning prediction distributions. Jia et al. (2019) and Yang et al. (2023) built noise-based and VAE-based external prediction obfuscator, respectively. Shejwalkar & Houmansadr (2021) tried to preserve stronger privacy via knowledge distillation on the extra training samples. Kaya & Dumitras (2021) studied the effort of various data augmentation on membership privacy. Jeong et al. (2022) tried to prevent attackers from data reconstruction via a generative noise injector. Guan et al. (2022) measured the risks of model stealing attacks via SAmple Correlation (SAC). Stadler et al. (2022) tried to analyze the effectiveness of generative data on privacy. Chen et al. (2022) controlled the prediction distribution of the training data within a specific interval via a piecewise objective function, RelaxLoss. Fang & Kim (2024) improved sample alignment in the relaxed state by adapting both RelaxLoss and CenterLoss.

**Generative Models**    Goodfellow et al. (2014) proposed adversarial generative nets to synthesize images. Mirza & Osindero (2014) tried to to generate conditional images. Radford et al. (2016) utilized transposed convolution in building GANs. Odena et al. (2017) introduced a classifier as an auxiliary training component to generate conditional images. Gong et al. (2019); Hou et al. (2022) further improved the auxiliary classifiers. Zhu et al. (2017) introduced CycleGAN into image collections transformation. Karras et al. (2020) studied how to improve image quality within limited training data. Casanova et al. (2021) added instances as auxiliary inputs to generate images similar to the instances. Ni et al. (2022) adopted manifold learning step into the discriminator. He et al. (2022) proposed a masked autoencoder to generate images according to broken clues. Hu et al. (2023a) studied the relationship between the latent space and data distributions. Hu et al. (2023b) compressed GANs via feature map distillation. Yu & Wang (2023) tried to build a generator from a pre-trained classifier directly. Wang et al. (2024) explored how generative models can help with contrastive learning. Ganjdanesh et al. (2024) proposed a GAN compression by enforcing structural similarity.

## 3 SAMPLE-LEVEL PRIVACY DATA VALUATION

With regard to the overall privacy of dataset, it is shown that the impact of models in Wang et al. (2021); Yuan & Zhang (2022); Tan et al. (2023); Chen et al. (2022) and data in Kaya & Dumitras (2021); Yu et al. (2021). The aggregate message is that privacy vulnerability is determined by a relative relationship between data complexity and model capacity. However, in these studies, it is shown that even if the privacy of a portion of the dataset is greatly enhanced, the improvement of the other part can be insignificant. That motivates us to ask two questions: i) Do distinctive privacy levels inherently exist among data samples? ii) May different models recognize the same samples as privacy-safe? We discuss these two questions in this section.

### 3.1 DO DISTINCTIVE PRIVACY LEVELS INHERENTLY EXIST AMONG DATA SAMPLES?

#### 3.1.1 WHEN DOES THE MODEL PREVENT DATA PRIVACY LEAKAGE?

For data valuation, each data sample needs to be relatively quantified and ranked in terms of resilience to privacy. We call such a ranking value as ***privacy level***. Since the privacy sensitivity depends on models, experimental settings, or each run, the privacy level is not an absolute value but a relative one. In order to properly distinguish and differentiate each sample from the others, the privacy levels of the samples need to be well separated. As we can see in Fig. 1a, when the overall

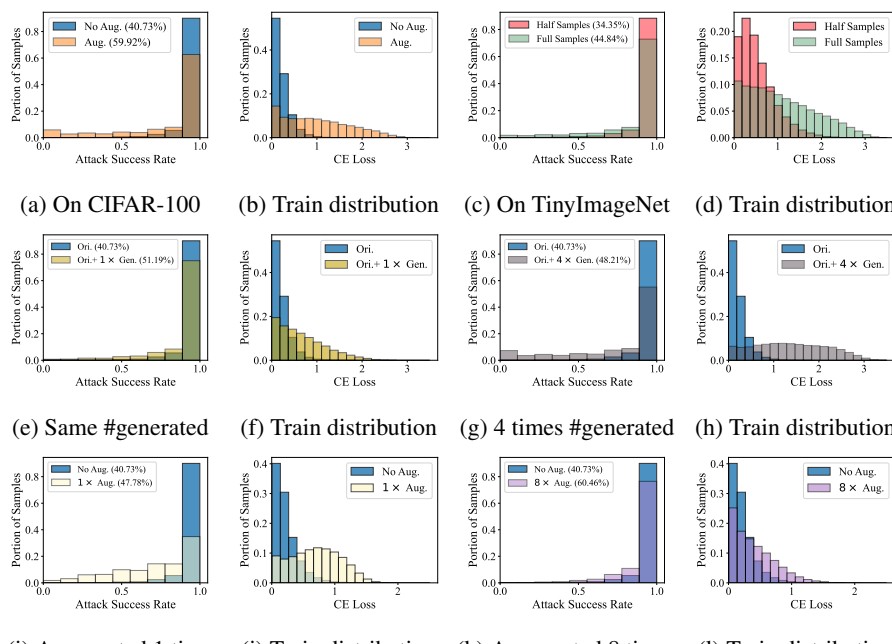

(a) On CIFAR-100    (b) Train distribution    (c) On TinyImageNet    (d) Train distribution

(e) Same #generated    (f) Train distribution    (g) 4 times #generated    (h) Train distribution

(i) Augmented 1 time    (j) Train distribution    (k) Augmented 8 times    (l) Train distribution

Figure 1: Per-example attack success rates and loss distribution on the original trainset (MobileNetV3-S, 40 independent runs). [The first row]: no augmentation vs. augmentation (left side); half vs. full samples (right side); [The second row]: original trainset only vs. with additional generated data; [The third row]: no augmentation vs. static augmentation. Testing accuracies parenthesized in legends. Attacks in these charts are M-Entropy Song & Mittal (2021).

privacy of the entire dataset changes, the impact/degree of the change is not even to every sample. This tells us that each sample inherently carries over different privacy traits, leading to proper separations between the privacy levels of the samples. Please note that existing studies focused on the overall privacy level of the model and the entire dataset but did not take into account *individual* samples' privacy levels.

Once the samples have proper gaps in their relative privacy levels, it is effective to categorize the samples into different privacy domains. To precisely and effectively enhance privacy safety, we eventually classify the samples into three **categories**:

- **Privacy-Safe**: a sample where privacy information is hardly leaked. It is resilient against privacy inference attacks.

- **Privacy-Risky**: it has the opposite property of privacy-safe samples, i.e., where privacy information is easily leaked and vulnerable to privacy inference attacks.

- **Privacy-Vague**: it can be either privacy-safe or privacy-risky depending on experimental environments/settings.

Unfortunately, such traits are hardly distinguishable by humans, which means that one cannot precisely identify privacy-safe samples manually. Another issue is that learning models with different architectures may prefer to memorize samples with different priorities or preferences. An aligned idea has been discussed in Ghorbani & Zou (2019); Wu et al. (2022). Memorizability and generalizability determine the "fitting" ability of the model to the training data and unseen data, respectively. Privacy risks mainly come from the discrepancy between the two characteristics. Memorizability is directly correlated with the model's capacity, while generalizability is not always. However, the decline in memory capacity does not always mean a decline in generalizability. For example, Frankle & Carbin (2019) shows that condensing the model with proper sparsity does not affect the generalizability.

**Observations & Conjectures**   To measure the privacy level of a sample, we calculate the *attack success rate*, $R = N_{hit}/N_{exp}$, where $N_{hit}$ denotes the number of times the sample is identified by an attacker, out of $N_{exp}$ number of independent experiments - the higher is more likely to be more vulnerable to attack (i.e., more privacy-risky), the lower more privacy-safe. The privacy level distribution of CIFAR-100 training samples is visualized in Fig. 1a & 1b. It is shown that more than half of the samples are vulnerable to the attacker, while there is still a portion of samples that are not easy to attack. However, once some augmentation techniques are applied, such as Random Flipping & Cropping Simonyan & Zisserman (2015) and Cutout DeVries & Taylor (2017) (implementation details are in Table. 1 in Appendix), a lot of privacy-risky samples are flipped into privacy-safer samples. (Please note that, in this paper, the notion of data *augmentation* means non-generative, simple augmentation approaches only.) An application of augmentation also drastically changes the loss distribution of the training (Fig. 1b) and testing accuracy (parenthesize in the legend of Fig. 1a). This addresses that the impact of samples on the model's memorizability and generalizability combi-natorially promotes privacy safety. In other words, the diversity of training samples can significantly enhance privacy safety by augmenting data samples. Conversely, considering such cropping/cutout augmentations, the removal of certain features on training samples, making them look like brand new instances to the model, can also promote privacy safety. Therefore, it brings us to two potential conjectures:

- **Conjecture i**: An excessive quantity of samples can force the model to choose only a limited number of samples to memorize,
- **Conjecture ii**: the model recognizes the samples with minor changes as new individual samples.

### 3.1.2   EXPLANATION ON CONJECTURES

**Conjecture i)**   First, to verify the first conjecture, we conducted two experiments: one is testing the impact of augmented samples (Fig. 1a & 1b); the other is comparing two different numbers of training samples of the same dataset (Fig. 1c &  1d). Fig. 1c exhibits that the double training samples challenge the memorization ability of the model (the half samples are sampled from the full samples with balanced quantities from all classes.) This result is consistent with one trained on CIFAR-100 with augmentation in Fig. 1a & 1b, indicating that privacy can be better protected by suppressing the memory capacity of the model with augmented samples. Tan et al. (2023) showed that increasing model capacity often hurts data privacy, and Stephenson et al. (2021) found the deeper layers usually leak more. Our experiment supports the observations and further reinforces this conjecture. However, the decrease of privacy risks in Fig. 1c (by doubling sample count) is not as significant as in Fig. 1a (by augmentation). One of the factors is that the model trained with half samples shows poor generalizability. The more important reason is the difference in the overall memorization capacity of the model for the actual training samples, which can be seen by comparing Figs. 1e vs.1g and Figs. 1i vs. 1k. Then, how can more samples help better privacy? To answer this question, we apply a generative model Gong et al. (2019) (details in Sec. A in Appendix) to increase the number of samples in CIFAR-100 to further verify our conjecture. Shown in Fig. 1e & 1f, the overall privacy of the model with real data has been improved after adding an equal quantity of fake data. When the training data quantity becomes four times in Fig. 1g, the model's privacy with real data is enhanced. Also, the CE loss distribution is more relaxed (flattened) in Fig. 1h than in Fig. 1f. These results are well aligned with conjecture i).

**Conjecture ii)**   For this conjecture to be true, if data augmentation was applied, a transformed sample should look unfamiliar or less familiar from the perspective of a model. In other words, the model should not memorize a real sample even by memorizing its augmented version. Looking at Fig. 1a (augmenting) and Fig. 1c (doubling sample count), data augmentation has a stronger impact than just doubling samples on privacy. One of the powerful pieces of evidence is in Kaya & Dumitras (2021). They addressed that not all augmentation techniques can help privacy, but Cutout, Random Cropping, and Label-Smoothing can. Cutout and Random Cropping are different from Label-smoothing in that they change the samples rather than training objects, by potentially removing some features. To verify it independent of conjecture i)'s effect, we produce the augmented data in the same vs. eight times quantity. Please note that, unlike Fig. 1a & 1b, this is augmented statically - for one time before the start of the train instead of being augmented at each epoch. Shown in Fig. 1i&1j, the model's memorization of real samples significantly decreases compared to

the training without augmentation. However, Fig. 1k&1l recovers the memorization of real samples, indicating that multiple versions of augmented samples but from the *same* real sample reinforce the model to memorize this real sample. Besides, data augmentation increases the difficulty of fitting to the data. Unlike real samples, for augmented samples, when the quantity of samples with data augmentation is insufficient, it is more difficult for the model to memorize the dataset, leading to better privacy but worse performance (shown in Table 2). Increasing the times of augmentations helps the model learn the data to achieve higher testing accuracy. However, it also poses a threat to real data privacy since the augmented samples contain a lot of clues about the real samples. Hence, unlike such static augmentations, dynamically renewing augmentation in every epoch (Fig. 1a & 1b) is more effective for the model to preserve privacy. This suggests that generative modeling will have more potential for privacy protection than augmented data. We discuss it in Sec. 4.

### 3.1.3 DATASET SIZE VS. COMPLEXITY

We have obtained collective insights and can answer the question: *Do distinctive privacy levels inherently exist among data samples?* The answer is *yes* and *no*. It is yes, when the dataset is complex enough and the model cannot afford all samples - then the model will be able to memorize only a small portion of the dataset. It is no, when model capacity outweighs data complexity so that the model is highly likely to memorize almost all samples, and from the model's perspective, all samples' privacy levels would be similar. This is also evident by referring to Carlini et al. (2022b) addressing that removing privacy-risky samples flips some other samples privacy-risky. Fortunately, however, such sample diversity can be significant under certain conditions that we can control. An electrical phenomenon, electrical breakdown Lehr & Ron (2018), can be an analogy to the certain conditions we show in the above experiments. With the analogy, we coin the term *data breakdown*,

- **Data Breakdown:** where a model becomes over or at capacity on data of high complexity so the model cannot memorize all but only a small portion of the training samples.

That is, until data breakdown occurs, a model cannot keep the data privacy well since it is able to memorize everything. Just more samples do not always contribute to increasing the data complexity. Only when the number of independent samples increases or the complexity of each/some of the samples increases does the data complexity increase and help privacy. Fig. 1c&1d prove that the increase of independent samples enhances privacy because the samples in the 1st and 2nd halves of the set are originally different (i.e., independent) samples. As opposed to that, comparing Figs. 1i&1j vs. 1k&1l shows that just 8 times more samples do not help privacy - it shows even a decrease over 1 time augmentation case. That is because the number of samples was increased by augmentation. Augmented samples are not independent but rather correlated because they are augmented from the same original samples. Please note that there exists a phenomenon called **data outlier effect** Choquette-Choo et al. (2021); Tramèr et al. (2022), which can easily be confused with **data breakdown**. To differentiate

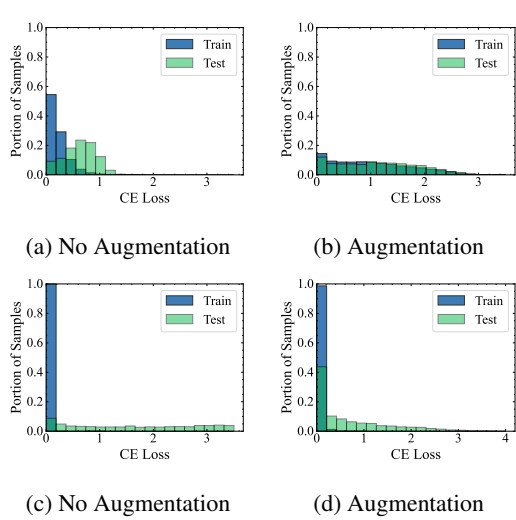

(a) No Augmentation          (b) Augmentation

(c) No Augmentation          (d) Augmentation

Figure 2: The CE loss distributions on Train and Test sets under various data capacities and model capacities. [The first row]: MobileNetV3-S; [The second row]: ResNet18.

them, we compare the prediction distributions with different model capacities at the same data quantity level. As shown in Fig. 2, the results on MobileNetV3-S show **data breakdown** while results on ResNet18 show **data outlier effect** only. When the data breakdown does not occur, the increase in data capacity improves the overall privacy level merely relies on generalizability improvement, which is the data outlier effect (please refer to Fig. 2c & Fig. 2d). In contrast, the prediction distributions on both the train and test sets change significantly *when the data capacity exceeds the model*

*capacity*, causing data breakdown (Fig. 2a & Fig. 2b). In a word, **data outlier effect** relies on the generalizability changes of the model while **data breakdown** relies on the relative changes between data capacity and model capacity. Fig. 1i exhibits that just a one-time augmentation shows better privacy than multiple augmentations (either dynamic (Fig. 1a) or static (Fig. 1k)), indicating that repeatedly augmented samples enforce a model to more memorize real train samples. This insight motivates us to generate more independent, distinguishable samples.

## 3.2 IDENTIFYING PRIVACY-SAFE AND -RISKY DATA

### 3.2.1 MOTIVATIONS AND CONDITIONS OF PRIVACY-LEVEL DATA VALUATION DESIGN

In Sec. 3.1, we show that increasing dataset complexity promotes privacy. Therefore, a direct and effective way for privacy is to generate fake data to increase the dataset complexity. However, it still does not precisely tell us which samples should be protected with a higher priority for more effective privacy protection. In other words, because the privacy levels are or need to be relative in a dataset, if there are no explicit privacy-risky samples, some portion of real data samples will become susceptible to privacy leakage. The real and fake data (generated but independent of the real data) are mixed among privacy-safe, -vague, and -risky portions. That is because, due to the memorization capacity, a model will always memorize a certain portion of samples fully, making it impossible to train a model with only privacy-safe data. That is, in an extreme case, even if a dataset is all composed of completely privacy-safe samples, the model will memorize a certain portion of samples, and then the samples become privacy-risky because they are memorized. In other words, the privacy of some samples must be sacrificed by memorization, even if they were privacy-safe in the first place. Therefore, with the necessity of diversity of privacy-safe/-risky samples in mind, the very ideal and extreme case is when all real (original) data are privacy-safe, while all fake (generated) data are privacy-risky, in order to protect the real data.

Then, a potential solution is to transform the data privacy category of the samples by transforming non-privacy-safe real data into privacy-safe data, while transforming non-privacy-risky fake data into privacy-risky data. We discuss how to determine privacy-safe/-risky samples. Firstly, a privacy-safe sample is supposed to have the following traits: i) A privacy-safe sample is unlikely to be recognizable by an attacker; ii) A privacy-safe sample is expected to consistently be identified as a privacy-safe sample even by various independently-trained models with various training techniques. Likewise, but in an opposite manner, we can also identify whether a sample is privacy-risky. Carlini et al. (2022b) proposed a way to determine whether a sample is easy to attack via dozens of independent model trainings and simulations of MIAs. In the work, however, if the training conditions are not sufficiently diversified, most samples can be incorrectly identified in repetition even in several independent experiments (e.g., Fig. 1a). For that reason, this approach can lead to unrealistic costs, although it would increase the training cost by tens of times. Having said that, their study revealed neural networks' two privacy issues: one is that different samples show different degrees of privacy risks. The other is that the relative privacy-risky degree of the data could be changed in different independent experiments, which may be caused by the randomness of the initialization of models, mini-batch selection, or training dynamics. That tells us that it is not feasible to determine a sample's privacy level just in a single or a few independent runs. With this insight in mind, we propose an approach to determine privacy-safe/-risky samples via less number of independent runs.

### 3.2.2 APPROACH

Consider a training set $D_{train}$ and a reference set $D_{ref}$. Threshold-based MIAs can be deployed based on them to a model trained on $D_{train}$. Next, suppose a threshold $\tau$ is built for MIAs to the model. Then the distance of each sample to $\tau$ can be calculated as $d(f(x), \tau)$, where $f(\cdot)$ is a forwarding function of the model and $d(\cdot, \cdot)$ is a distance measuring function. The distance function $d(\cdot, \cdot)$ is used as a metric to measure how privacy-safe a sample is. If the closer the sample's prediction $f(x)$ is to the $\tau$, the safer it is because the attacker's judgment on it would be more uncertain.

However, the relative memorization characteristics of the model and randomness in training techniques inevitably cause inconsistent levels of privacy safety of samples across multiple independent experiments. Hence, we identify overlapping (common) privacy-safe samples across multiple runs. The *overlap ratio* is a metric we propose to represent the agreement degree of multiple independent

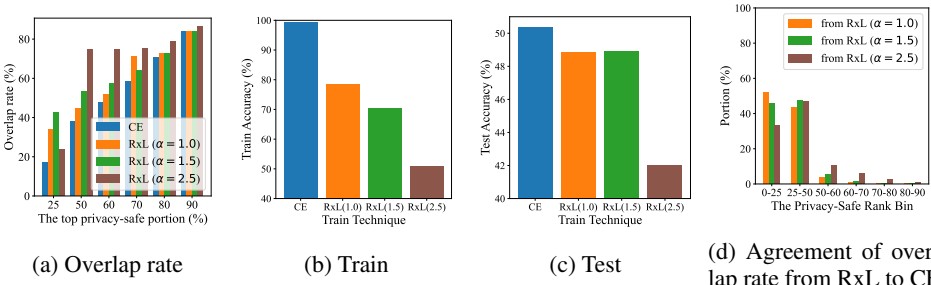

(a) Overlap rate      (b) Train      (c) Test      (d) Agreement of overlap rate from RxL to CE

Figure 3: (a) The overlap rate of privacy-safe samples (b) the training accuracy (c) testing accuracy (d) the agreement percentage of the original rank of top 50% common privacy-safe samples of the RxL-based models when they were CE-based models. (MobileNetV3-S, CIFAR-100, over five runs)

runs. Once the samples' privacy levels are ranked, we can selectively consider samples within a certain ranking bin. For example, we can consider samples in the top 20%, and the ranking bin will be represented as $[0\%, 20\%]$. If we would consider samples in an intermediate range, such as the top 60% but excluding the top 50%, the bin will be $[50\%, 60\%]$. We denote the starting and ending ratio as $b_s$ and $b_e$, respectively, and the bin will be represented as $[b_s\%, b_e\%]$. For such a certain privacy ranking bin, the overlap ratio is defined as follows,

$$R_{overlap} = N_{com}/N_{bin} \tag{1}$$

where $N_{com}$ denotes the number of common samples in the privacy rank bin among all runs, and $N_{bin}$ denotes the maximum number of samples that can be common in the bin. When $N_{total}$ is the number of all training samples, $N_{bin} = N_{total} \times (b_e\% - b_s\%)$. Fig. 3a displays the overlap ratio of the top-ranked privacy-safe samples over five runs. We observe that, for a Cross-Entropy-based (denoted as CE) and fully trained model, the agreement of sample rankings on independent experiments is low, especially for the top 25% privacy-safe samples. Hence, we investigate the samples' privacy-safe rankings also in defense models, RelaxLoss, Chen et al. (2022) (denoted as RxL). Compared to CE, in RxL, we can check that the agreement in overlap rate of top 25% and 50% privacy-safe samples significantly increases. The train and test accuracies are shown in Fig. 3b and 3c, respectively. This explains that RelaxLoss always gives priority to protecting some samples with specific characteristics. On the other hand, a model tends to remember certain samples containing certain features.

Then, one might ask this question: are the privacy-safe samples that are determined by CE also determined as privacy-safe by RxL? To answer this question, we investigate if the privacy-safe samples identified by RxL are also identified by the CE-based model. In Fig. 3d, we plot the top 50% common privacy-safe samples found via RxL-based models, which were identified by the CE-based model. Most of the top 50% privacy-safe samples' average ranking in the CE-based model is still in the top 50% portion in RxL. That tells us that these samples are decidedly privacy-safe as they are identified consistently by multiple models.

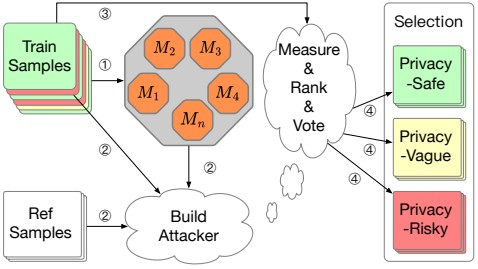

Figure 4: The illustration of privacy-level sample valuation and categorization.

The process of privacy-safe sample determination is shown in Fig. 4. We train multiple models via defense mechanisms (such as RxL) from scratch or the CE-trained approach to arrange an ensemble. Then, we measure and rank the samples via an attack technique. Finally, the ensemble votes for the most privacy-safe and -risky samples in the training set.

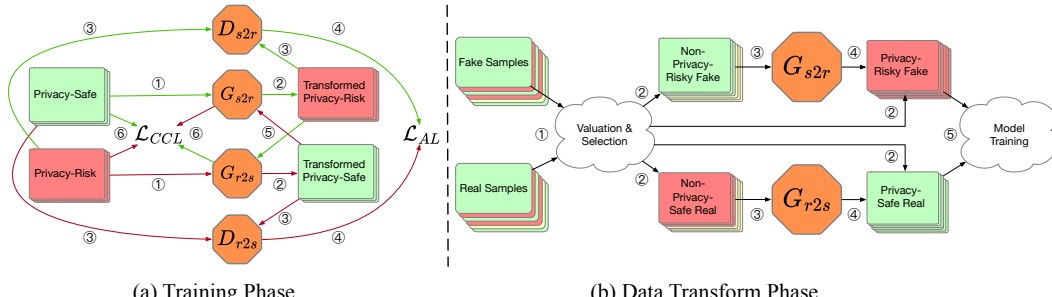

(a) Training Phase  (b) Data Transform Phase

Figure 6: The illustration of privacy domain transformation and model training. In (a), the green line represents the workflow from privacy-*safe* samples, while the red line is for privacy-*risky* samples.

## 4 DATA GENERATION

**Privacy Domain Transformation by Data Generation** The different privacy domains of training samples can be measured and evaluated, so they are potentially learnable. Hence, a generative approach is designed for privacy domain transformation. Suppose that we have determined samples in two domains, the privacy-safe domain, $\Omega_s$, and privacy-risky domain, $\Omega_r$. To transform data from each of these two domains, two different generators $G_{s2r}$ and $G_{r2s}$ are deployed. The generator $G_{s2r}$ transforms samples from $\Omega_s$ into new samples to be indistinguishable by $\Omega_r$, and vice versa for $G_{r2s}$, $\Omega_r$, and $\Omega_s$.

Although removing more information from the real data will ideally help more privacy, the loss of high-quality features from real data would be crucial to the performance of the model. Moreover, high-quality fake samples also have a potential risk of privacy leakage since these high-quality features are likely to contain similar features from real data. As shown in Fig. 6b, we try to transform all non-privacy-risky fake data to the privacy-risky domain $\Omega_r$, and all non-privacy-safe real data to the privacy-safe domain $\Omega_s$. To enable this, the domain transformation generative technique is designed. When training the privacy-preserving model, we empirically found that symmetrical data augmentation on both real and fake data is

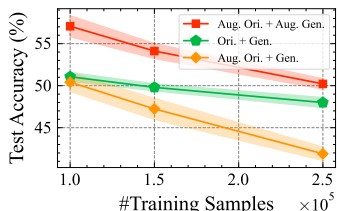

Figure 5: Comparison among different augmentation options

better for the model's generalizability (shown in Fig. 5). For a generative model, the CycleGAN architecture Zhu et al. (2017) is utilized to transform data privacy domains.

**Training Generators** For training generators $G_{s2r}$ and $G_{r2s}$, two corresponding discriminators, we also employ $D_{s2r}$ and $D_{r2s}$ in CycleGAN's training phase. The generator $G_{s2r}$ transforms privacy-safe samples, $I_s \in \Omega_s$, into privacy-risky samples, $G_{s2r}(I_s)$. The discriminator, $D_{s2r}$, cannot identify $G_{s2r}(I_s) \notin \Omega_r$ after adversarial training dynamics. Likewise, $D_{r2s}$ cannot distinguish $I_r \in \Omega_r$ and $G_{r2s}(I_r)$. The overview of the training phase is shown in Fig. 6. The adversarial loss, $\mathcal{L}_{AL}$, Goodfellow et al. (2014) is formulated as follows:

$$\begin{aligned}
\mathcal{L}_{AL}(G_{s2r}, G_{r2s}, D_{s2r}, D_{r2s}) &= (D_{s2r}(G_{s2r}(I_s)) - 1)^2 + D_{s2r}(I_s)^2 \\
&\quad + (D_{r2s}(G_{r2s}(I_r)) - 1)^2 + D_{r2s}(I_r)^2
\end{aligned} \tag{2}$$

Besides that, to ensure the transformed sample can be aligned with the original sample as much as possible, the cycle consistency loss, $\mathcal{L}_{CCL}$, Zhou et al. (2016) is introduced to keep the informative consistency between the original sample and the transformed sample. It is formulated as,

$$\mathcal{L}_{CCL}(G_{s2r}, G_{r2s}) = \|G_{r2s}(G_{s2r}(I_s)) - I_s\|_1 + \|G_{s2r}(G_{r2s}(I_r)) - I_r\|_1 \tag{3}$$

As a result, the total loss for training CycleGAN, $\mathcal{L}_{CycleGAN}$, is formulated as,

$$\mathcal{L}_{CycleGAN}(G_{s2r}, G_{r2s}, D_{s2r}, D_{r2s}) = \mathcal{L}_{AL}(G_{s2r}, G_{r2s}, D_{s2r}, D_{r2s}) + \lambda \mathcal{L}_{CCL}(G_{s2r}, G_{r2s}) \tag{4}$$

where $\lambda$ is a coefficient to balance these two losses. With these losses, the generators can properly learn how to convert samples between privacy-safe and -risky domains with unpaired sample training.

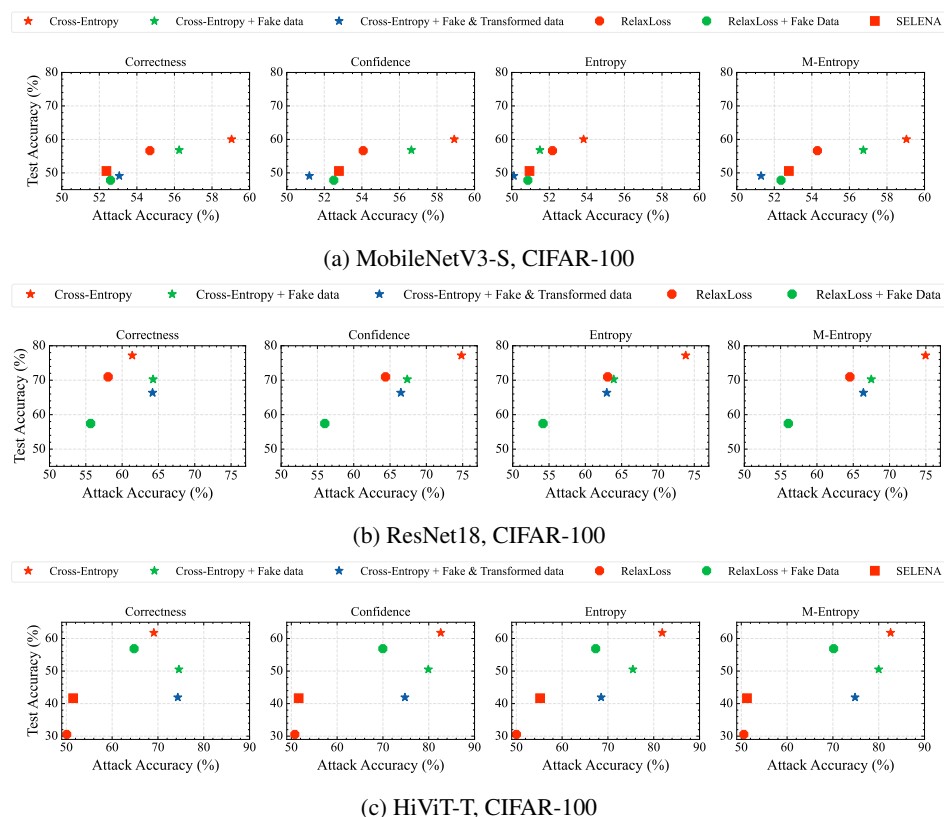

Figure 7: Generalizability and Privacy Performance with MobileNetV3, ResNet, and HiViT on CIFAR-100 and TinyImageNet. Additional results are included in Appendix due to space limitations.

## 5 EMPIRICAL VALIDATION

### 5.1 EXPERIMENTAL SETTINGS

We evaluate our approach on CIFAR-100 Krizhevsky et al. (2009) and TinyImageNet Le & Yang (2015). On CIFAR-100, we evaluate our approach with MobileNetV3 Howard et al. (2019), ResNet18 He et al. (2016), and HiViT-T Zhang et al. (2023) (a transformer model). On TinyImageNet, we evaluate our approach with MobileNetV3 and ResNet18. The SGD and Adam optimizer are applied to train CNN and transformer models, respectively. In training on both datasets, for augmentation, Random Cropping and Random Flip are applied for all cases. For privacy attacks, we evaluate approaches on correctness-based attacks (`Correctness`) Yeom et al. (2020), confidence-based attacks (`Confidence`) Song et al. (2019), entropy-based attacks Shokri et al. (2017) (`Entropy`), and modified entropy-based attacks (`M-Entropy`) Song & Mittal (2021). For comparison, SELENA Tang et al. (2022) and RelaxLoss Chen et al. (2022) are also evaluated. Every experiment presented in this paper takes at most a day to run with the average task requiring only a few hours on NVIDIA RTX4060Ti or A100 GPU.

### 5.2 RESULTS AND DISCUSSION

As shown in Fig. 7, we study the performance regarding testset and defending MIAs among different approaches and training data. Among all four MIAs, the most effective attacks are correctness-based MIAs and M-Entropy MIAs. Training models are with only real trainset, and cross-entropy is the most privacy-risky case in Fig. 7a. Training with additional fake data and transformed data has advantages in defending confidence-, entropy-, m-entropy-based MIAs compared with RelaxLoss and SELENA. As for CE and fake data, CE with fake and transformed data has better privacy but

worse testing accuracy, showing that our approach clearly changes the model's memorizing priority on training samples (although a part of the further decrease in testing accuracy is caused by the poorer quality of transformed samples.) In contrast, RelaxLoss and SELENA are better at defending correctness-based MIAs. This is because they set the maximum fitting degree of the model on samples, mini-batch level, and sample level as a training object.

In Fig. 7b, a change in setting is that the model capacity is much higher, leading to total privacy being more risky. Also, it shows better results in terms of generalizability. Under this condition, we find that the Correctness-based MIAs become more risky when we train the model using CE with the $1\times$ quantity of fake data or fake & transformed data. Note that the result of SELENA is not displayed since ResNet18 frequently suffered from crushes when it was trained with SELENA, which is worth noting.

In Fig. 7c, the trends change a lot. Due to the soft weights, the HiViT memorizes the training data deeply. Compared with other training techniques, the model shows deeper memorization on the trainset. One difference is that the model shows better test accuracy when training with RelaxLoss and fake data. This is because the data complexity requirement of the transformer model is usually higher than CNNs. Although the fake data's quality is not improved, the model can still learn the weights better with them.

In Fig. 12 (in Appendix), we find that there are some exceptional cases when the fake data does not help privacy. That is because the generator always produces similar but poor-quality (too noisy and distorted) samples based on TinyImageNet (shown in Fig. 11b). Hence, it is important to generate fake data with diversity while maintaining generalizability as much as possible.

Additionally, we evaluate MobileNetV3-S, trained on CIFAR-100, with Likelihood Ratio Attack (LiRA) Carlini et al. (2022a). Shown in Fig. 8, it can be seen that the

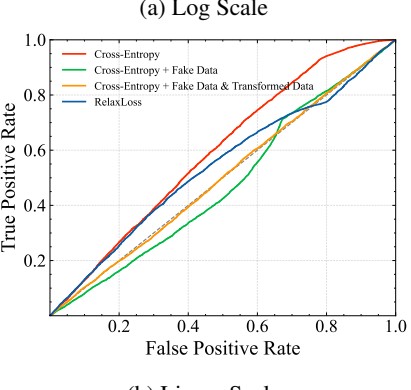

(a) Log Scale

(b) Linear Scale

Figure 8: Evaluations on LiRA.

model trained with fake data and transformed data is still effective on privacy. When the model is trained with both transformed data and fake data, the ROC curve becomes closer to the gray dashed line, indicating that our approach is effective against MIAs.

## 6 CONCLUSION

In this paper, we observed privacy vulnerability caused by asymmetry of data complexity and model capacity. The fundamentals of the solution are to prevent the model from memorizing more samples. That is a process to transform samples privacy-safe. To achieve the goal, we conducted sample-level data valuation, unlike existing literature, to categorize them as privacy-safe or -risky. On top of that, we generated more fake samples and ultimately transformed the samples' privacy domain. Through extensive empirical results, we showed our approach is significantly capable of defending privacy attack attempts.

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

APPENDIX

## A   MORE DETAILS FOR SEC. 3.1

Table 1: The data augmentation deployed on CIFAR-100 in Sec. 3.1

| Data Aug. Tech. | Conf. | Occur Probability |
|---|---|---|
| Random Cropping | size=$32 \times 32$, padding=4 | 1.00 |
| Random Flip | Horizontal Only | 1.00 |

**Configurations on data augmentation**   The data augmentation details of CIFAR-100 experiments are shown in Table 1. To make the augmented samples as independent as possible from the original samples for privacy safety, we set the occurrence probability of all data augmentation techniques to 100% (in fact, data augmentation on CIFAR-100 in most common experiments is also set to 100%.)

**Generator for generated data**   To synthesize data, we employ a generative model. In our study, Twin Auxiliary Classifiers GAN (TAC-GAN) Gong et al. (2019) is utilized. There are several reasons to use it:

- It can directly generate data of the corresponding class we need, which makes the experiments convenient.
- GAN is widely used in literature and practice. Hence it is good for reproducibility.
- Compared with other generative techniques (e.g., VAE, Flow-Based Models, and Diffusion Models), GAN is rich in generative diversity.
- We find that generating high-quality data across various datasets is challenging. Fortunately, our method does not require generated data to contribute to the generalizability of the model, indicating our approach is highly adaptable.

Table 2: The average training information among 40 runs. (CIFAR-100, MobileNetV3-S)

| Train data | Train data acc. (%) | Train data loss | Real train data acc. (%) | Real train data loss |
|---|---|---|---|---|
| No Aug. | n/a | n/a | 99.50 | 0.06 |
| Aug. | 72.22 | 0.92 | 77.43 | 0.76 |
| $1\times$ Aug. | 92.85 | 0.27 | 57.89 | 1.86 |
| $8\times$ Aug. | 94.74 | 0.22 | 97.76 | 0.14 |

## B   MORE DISCUSSION FOR SEC. 3.2

### B.1   WHY NOT MERELY TRAIN WITH GENERATED DATA

The main reason we do not directly use generated data to train the model for real data privacy is that the quality of generated data is hard to compete with natural data (real data), which can be seen in Fig. 10 & 11. For generative models, extracting more generalized features from real data is risky in terms of privacy. On the one hand, the model may be overfitted, causing the generated data to lose both diversity and independence, leading to the generated data itself also becoming risky to real data privacy. As shown in Fig. 9, the model trained with only generated data shows very poor generalizability even with much more samples than original data (real data). On the other hand, data-free learning is a challenging issue, especially on challenging datasets, which makes training with generated data unrealistic. In contrast, the model trained with both original and generated data shows acceptable generalizability (it sometimes shows even better testing accuracy than the model trained with real data only.) Therefore, it is worthwhile to consider how to protect real data privacy better, given the training model with both real data and generated data.

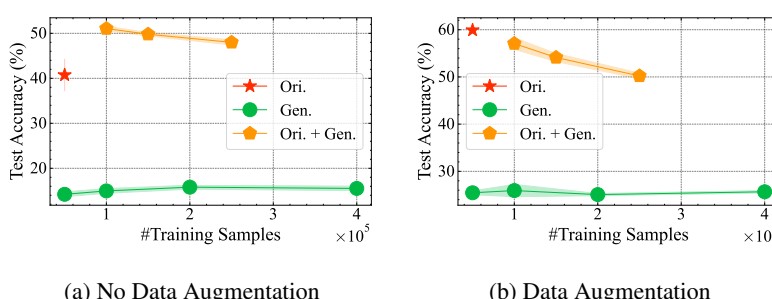

(a) No Data Augmentation      (b) Data Augmentation

Figure 9: Performance comparison between training with original (real) trainset and different size of generated trainset.

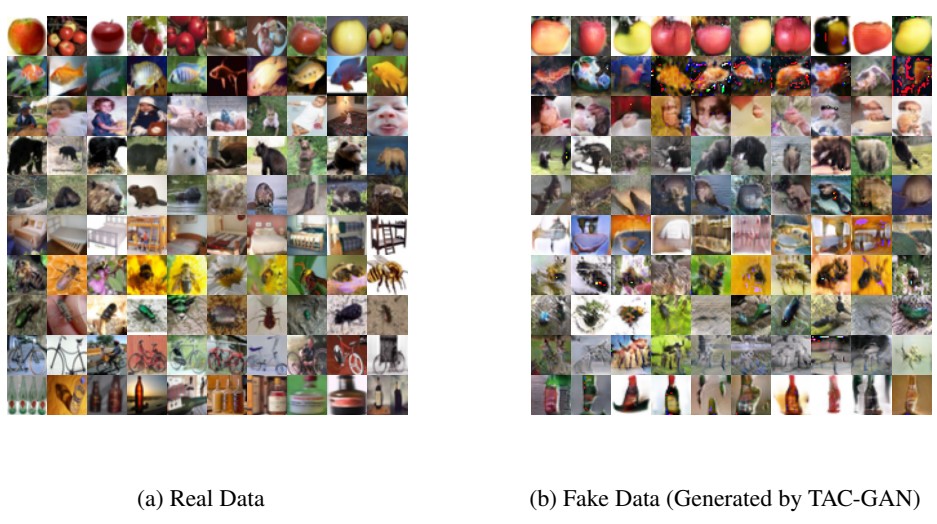

(a) Real Data        (b) Fake Data (Generated by TAC-GAN)

Figure 10: Samples of real data and fake data (generated) on CIFAR-100

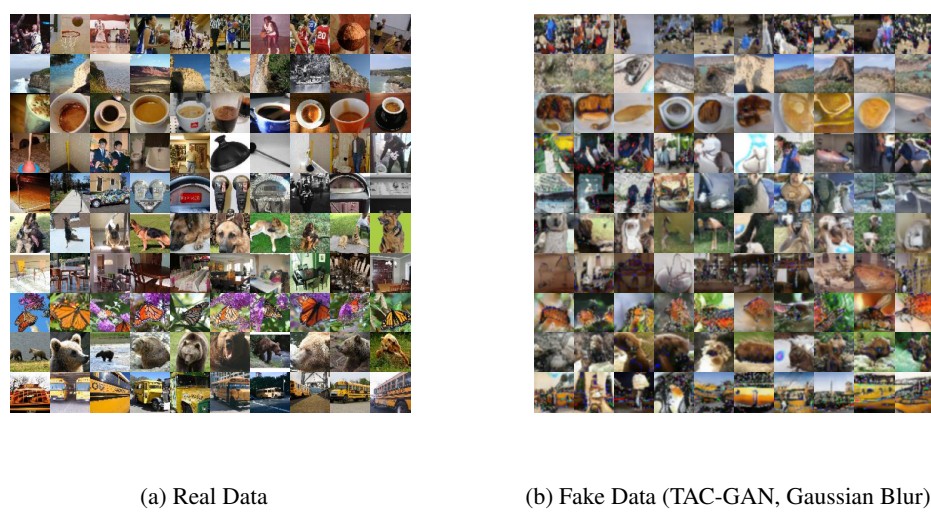

(a) Real Data        (b) Fake Data (TAC-GAN, Gaussian Blur)

Figure 11: Samples of real data and fake data (generated) on TinyImageNet

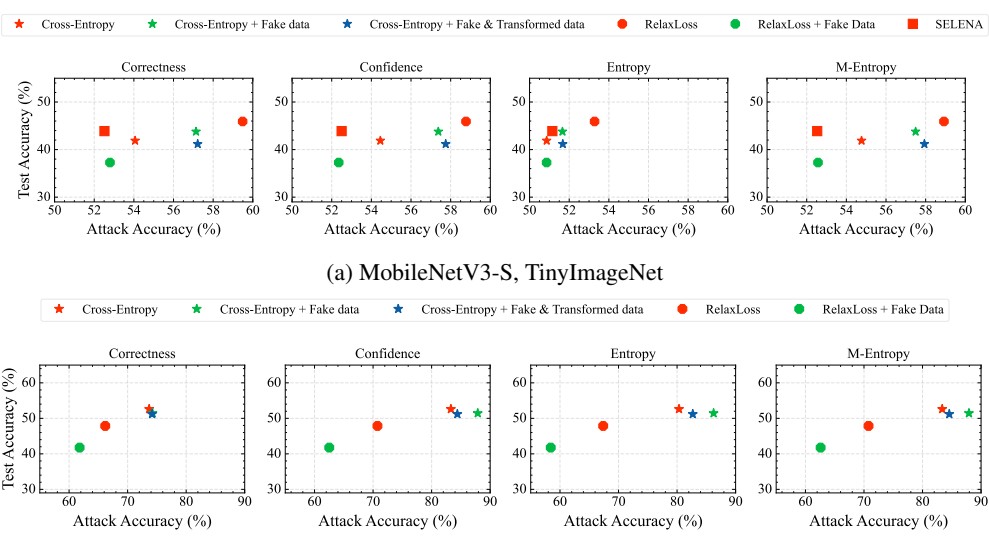

(a) MobileNetV3-S, TinyImageNet

(b) ResNet18, TinyImageNet

Figure 12: Additional results for Fig. 7

