# OpenReview forum: "Generating Fake Data to Fake Privacy Pryers"
_ICLR.cc/2025/Conference — ICLR 2025 Conference Withdrawn Submission_

### Official Review · Reviewer_pfF9 · 2024-11-03

**Soundness:** 2
**Presentation:** 2
**Contribution:** 1
**Rating:** 3
**Confidence:** 4

**Summary:**

The authors posit that data asymmetry is prone to privacy violations in the context of membership inference attacks. The manuscript proposes a generative process that makes use of CycleGANs to transform data samples (symmetrically) between privacy-safe and privacy-risky domains, judged using an MI-attack against an ensemble of models trained with pre-existing defense mechanisms. The results show that augmenting the training set using this process leads to better privacy-utility tradeoffs in some circumstances.

**Strengths:**

- The authors do a good job at establishing the motivation behind their approach incrementally, starting with the data distribution.

- The application of CycleGAN to transform data between privacy-safe and privacy-risky domains is quite creative.

**Weaknesses:**

- The related work section does not sufficiently contextualize the manuscript with respect to prior work - the section is a list of one-line summaries of papers, but it is not made clear to the reader why many of those are directly relevant.

- The writing throughout the paper needs significant refinement - not grammatically, but often the point the authors are trying to make is entirely unclear at first. For instance, Section 3 begins with what seems like an incomplete sentence. Another example is where the authors start talking about the results in Fig. 1 without offering sufficient detail about what is being presented. The fact that the 'attack success' is based on the M-Entropy attack is tucked away in the image caption. What it means for "the privacy of the entire dataset" to change, and how privacy is quantified in the study, is not addressed until later, which was very frustrating. Similarly, a short summary of RelaxLoss would have been aided readability.

- My main concern with the manuscript is a general lack of precision in the writing. For instance, a claim such as "This explains that Relax Loss always gives priority to protecting some samples with specific characteristics. On the other hand, a model tends to remember certain samples containing certain features." is not informative. Which specific characteristics, and how are they quantified? Which samples, and why?

**Questions:**

- Section Conjecture i): How do Fig 1. a-b support the observation/conjecture that model complexity hurts privacy, and deeper layers leak more information? If that is not what was meant, I don't see why the references to Tan et al. (2023) and Stephenson et al. (2021) are relevant here. We have no information about the model architecture used at this point.

- How is the data outlier effect defined? Neither reference contains a concrete definition - so I'm assuming the authors of this manuscript are referring to the effects of data outliers on privacy at large - in which case, it would be useful to define it in the context of the proposed work, in order for the reader to interpret and evaluate the results in Fig. 2 with sufficient information. Currently, it is unclear to me why the ResNet18 plot demonstrates the data outlier effect.

- How many (and which) models make up your ensemble? What is the voting mechanism? Did you use the M-entropy attack here?

---

### Official Review · Reviewer_9AtV · 2024-11-04

**Soundness:** 1
**Presentation:** 1
**Contribution:** 1
**Rating:** 3
**Confidence:** 4

**Summary:**

This paper studies how to enhance data privacy by generating fake data. The paper investigates privacy level for each data sample based on attack success rate, which is an empirical indicator defined by the ratio of success hits. Based the attack success rate, the paper identifies three privacy levels: privacy-vague, privacy-safe, and privacy-risky. The proposed approach uses GAN to construct a process called transformation, which generates fake data samples (from privacy-safe to privacy-risky and from privacy-risky to privacy-safe). Numerical experiments are provided to show the performance of the proposed approach.

**Strengths:**

This paper captures a very interesting angle to generate fake data to enhance data privacy by leveraging the individual privacy risk of the data samples.

**Weaknesses:**

This paper has several significant issues and flaws. The presentation of the paper is poor. The use of the English language needs to be improved. In addition, there are no rigorous theoretical and analytical results, and many claims and statements are speculative. The validity and usefulness of the proposed approach is questionable.

Comment 1:
It seems that only 6-10 pages focus on the proposed approach. There are many of unnecessary content on the first 5 pages. In addition, many parts have logic flaws; there are statements or claims that are presented based on motivations of irrelevant information. A few examples are shown as follows.

In the paragraph Observations & Conjectures, the statement “In other words, the diversity of training samples can significantly enhance privacy safety by augmenting data samples” does not logically follow from the previous observations. The preceding discussion is about specific augmentation techniques (like flipping and cropping) that may reduce privacy risks for some samples, but it does not establish a general principle that diversity alone inherently enhances privacy. Terms like “data complexity,” “model capacity,” and “privacy safety” are used with almost no explanation or context. The reader is left without a clear understanding of what constitutes “enough complexity” or how “model capacity” specifically impacts privacy.

Section 3.1.2, which explains the two conjectures, has issues in terms of clarity, logic, and depth. The explanations for both conjectures lack empirical or theoretical support. Conjecture i claims that a larger sample quantity forces the model to memorize fewer samples selectively. However, there’s no direct evidence or formal analysis to substantiate this assumption. Conjecture ii suggests that minor transformations lead the model to see samples as “new,” thus reducing memorization. However, there is no experimental validation or reference to prior work showing that minor augmentations significantly impact memorization in this way. The key terms such as “model’s memory capacity,” “data complexity,” and “unfamiliar samples” are used without clear definitions or context. This leaves their meanings open to interpretation. For instance, the section refers to “challenging memorization,” but it’s unclear what specifically constitutes a “challenge” for the model in this context. The section does not effectively explain the causal relationships implied by the conjectures. For example, while the section suggests that increased sample diversity or transformations make memorization more difficult, it does not provide a clear, step-by-step rationale for how or why this is the case. The explanations are heavily speculative, presenting assumptions about how models behave without rigorous testing or supporting arguments. Conjectures are presented as if they are established insights, but the lack of verification weakens the overall argument and may lead readers to question the validity of these explanations.

In addition, in Section 3.1.3 DATASET SIZE VS. COMPLEXITY, the answers provided for “yes” and “no” in the paper are problematic. The answers given for both “yes” and “no” lack general applicability. The idea that a model will only memorize a “small portion of the dataset” when the dataset is complex and the model capacity is limited, or that it will memorize “almost all samples” when capacity outweighs complexity, is an oversimplification. In practice, model memorization patterns are influenced by a variety of factors, including sample distribution, regularization techniques, and specific characteristics of individual samples (e.g., outliers or unique patterns), not just dataset complexity and model capacity. The answers ignore these factors and provide an overly simplistic view. The paper does not provide any empirical evidence, theoretical proofs, or references to prior work that support these claims. In research, assertions like these would typically be backed by experimental results, existing literature, or formal proofs. Without support, these statements come across as conjectural and speculative rather than rigorous findings, which weakens the paper’s credibility and makes it challenging for readers to trust these conclusions. The question-and-answer format attempts to address a concept (sample-level privacy under different conditions) in a way that’s both inconsistent and overly simplistic. For example, even if model capacity did influence memorization, this wouldn’t inherently mean that privacy levels across samples become uniform or distinct. This binary reasoning ignores the subtlety of model-data interactions and leads to potentially misleading conclusions.

The above are just a few examples of the flaws of this paper.

Comment 2:

As described on page 4, "However, once some augmentation techniques are applied, such as Random Flipping & Cropping Simonyan & Zisserman (2015) and Cutout DeVries & Taylor (2017) (implementation details are in Table. 1 in Appendix), a lot of privacy-risky samples are flipped into privacy-safer samples. (Please note that, in this paper, the notion of data augmentation means non-generative, simple augmentation approaches only.) An application of augmentation also drastically changes the loss distribution of the training (Fig. 1b) and testing accuracy (parenthesize in the legend of Fig. 1a).", the augmentation techniques change the privacy level of data samples. Hence, I have a question:

Q1: Does the involvement of the fake data generated by the proposed approach effectively change the privacy level?

That is, my concern is that the privacy levels of the true data samples and the privacy levels of the generated fake data samples (which are transformed to have a certain privacy level based on the privacy level of the true data samples) would be different when the new input dataset is input to the model.

The attack success rate is an empirical black-box evaluation of individual privacy level, which is a joint consequence of all input data samples (not just the specific individual data sample) and the underlying model (as also pointed out by the paper, e.g., "Another issue is that learning models with different architectures may prefer to memorize samples with different priorities or preferences.", which leads to different privacy levels). Hence, the individual privacy level of a data sample is in general sensitive to the "environment" (i.e., all other input data samples and the model) of the data sample.

Therefore, the transformation performed by the proposed approach may not actually transform the fake data samples to the anticipated privacy level.

Comment 3:

The proposed approach uses a GAN-fashion generative framework given the privacy-safe domain $\Omega$\_$s$ and the privacy-risky domain $\Omega$\_$r$. By the nature of GAN, the proposed approach implicitly requires the assumption that there are underlying well-defined probability distributions for the privacy-safe and the privacy-risky fake data samples. After the training, the discriminator is supposed not to distinguish the true sample in $\Omega$\_$s$ or $\Omega$\_$r$ from the generated fake data samples, respectively. Therefore, ideally, the "target" distributions of the generated fake data samples should be very close to the true distributions of the samples in the sets $\Omega$\_$s$ and $\Omega$\_$r$.

However, as pointed out above, the privacy levels of the (true and fake) samples are in general very sensitive to all input data samples and the model. Therefore, the distributions of the fake samples (and hence the distributions of the samples in $\Omega$\_$s$ and $\Omega$\_$r$)  are also in general very sensitive to the input data samples and the model. (Let me know if I am missing something.)

In general, random subsets of a dataset with a fixed distribution are expected to follow the same overall distribution as the original dataset. However, specific subsets defined by properties related to model behavior, such as memorization likelihood or attacker success rate, might possibly exhibit distributional shifts, which is in general a rare phenomenon and requires restrictive treatments. Without rigorous theoretical proofs, it is hard to believe that the proposed approach is indeed applicable.

There is another issue with the GAN-based framework. In the standard GAN, the input of the generator is a random variable (noise) with a given distribution. However, in the proposed approach, the input to the generator is a true data sample. By using a true data sample as input, the generator is constrained to transform that specific sample, which limits the diversity (and randomness) of the generated outputs. This could lead to repetitive or overly similar transformed samples. Therefore, there is a risk that the generator might overfit to certain patterns within these samples, which potentially carries over identifiable features from the original data samples. This could undermine privacy.

The authors should use more space to clearly and rigorously describe their proposed approaches.


Comment 4:

The experiments in the paper compare the proposed approach exclusively with “Random Cropping and Random Flip” (though these techniques do not appear explicitly in the plots). This comparison is limited, as it fails to include a broader range of baseline methods that would provide a more comprehensive evaluation. Given the numerous techniques discussed in the first five pages, incorporating additional baselines would strengthen the experimental analysis and offer a clearer picture of the proposed method’s effectiveness relative to established approaches.

Q2: In the experiments, is "fake data" also generated by the proposed approach?

Q3: What is the difference between "fake data" and "fake data & transformed data"?

Q4: Moreover, the plots do not include Random Cropping and Random Flip. Are the "fake data" generated by Random Cropping and Random Flip?

Q5: Does "fake data & transformed data" include both "fake data" and "transformed data"?

Q6: In addition, the authors should clearly describe the number of input data samples for each experiment. Are the numbers of input data samples for "fake data"  and "fake data & transformed data" the same in the experiments?

**Questions:**

See Weaknesses.

---

### Official Review · Reviewer_XWpV · 2024-11-04

**Soundness:** 2
**Presentation:** 3
**Contribution:** 3
**Rating:** 5
**Confidence:** 3

**Summary:**

The authors propose a method to empirically measure privacy at the sample-level, enabling them to categorise samples into different privacy levels and then generate fake data from these categories. This synthetic data is then used to obfuscate real samples during the training of the model and guarantee a form of empirical privacy.

**Strengths:**

- The approach to using fake data as a way to guarantee a form of empirical privacy is fairly novel.
- The empirical insights for sample-wise privacy-levels is interesting and of wider interest.
- The paper is well-written and clear.

**Weaknesses:**

- The overall effectiveness of the method seems unclear from the provided experiments in terms of the privacy-utility of the end model.
- It is not clear why you should use the proposed approach over the considered baselines (SELENA/RelaxLoss).
- More could be done to explore how the proposed method compares with formal privacy e.g., Differential Privacy.

**Questions:**

- In Figure 7, there are certain settings where the proposed method is not effective e.g., correctness-based attacks on ResNet18 CIFAR100 where the standard cross-entropy model is actually better than the proposed approach. Why do you think this is case? Is it related to certain cases mentioned in Fig 12 in the Appendix?
- Related to the above, what is the main advantage of using the proposed fake data approach? Would you say that the fake data approach allows for a better balance of accuracy and attack resistance when compared to SELENA/RelaxLoss over a wider range of different attacks? This is not immediately clear to me from Figure 7.
- With regards to the baseline methods, is not clear why you should use the proposed approach compared to SELENA/RelaxLoss as there are many scenarios where SELENA seems to give better models.
- Is there any significant overhead to generating the fake data for training? How does this compare to the baselines?
- It could help to have an algorithm clearly outlining the fake data approach, as the method is defined over multiple sections and it is easy for a reader to lose track.
- Lines 497-500, could you clarify what is meant by ResNet18 suffering “crushes” when trained with SELENA?

---

### Official Review · Reviewer_AujC · 2024-11-09

**Soundness:** 2
**Presentation:** 3
**Contribution:** 2
**Rating:** 5
**Confidence:** 3

**Summary:**

This paper presents a new approach to mitigating privacy vulnerabilities in machine learning models.
The core idea is to generate more data samples to prevent models from memorizing individual data points.
The authors propose a generative method that creates privacy-risky fake samples, which confuse attackers and protect the original, privacy-safe data.
The approach focuses on analyzing each individual data sample's privacy level rather than evaluating the dataset as a whole.
By categorizing data samples into privacy-safe, privacy-vague, and privacy-risky categories, they aim to protect the privacy of real data while generating fake data that attackers might target.
The method uses data transformation and generation techniques to adjust the privacy levels of these samples.

**Strengths:**

(+) Generating privacy-risky fake data to confuse attackers seems an innovative and elegant solution for privacy-preserving ML.

(+) The paper provides a good methodology, including the categorization of data into privacy-safe, privacy-vague, and privacy-risky categories. The approach is rigorously validated through extensive empirical evaluations using popular datasets like CIFAR-100 and TinyImageNet, and it compares favorably against established defenses like RelaxLoss and SELENA.

(+) The paper clearly explains the motivation behind the proposed approach (with two conjectures), making the reader understand why privacy risks arise due to data and model asymmetry and how generating privacy-risky fake data can mitigate these risks.

**Weaknesses:**

(-) The experiments are limited to image datasets (CIFAR-100 and TinyImageNet), but privacy risks are context-dependent and can vary significantly across other types of data, such as text or tabular data.

(-) The privacy-utility trade-off is not sufficiently addressed or quantified, especially when transformed samples are introduced. A more in-depth discussion and quantification of this trade-off, along with potential methods to mitigate it, would enhance the quality of the paper.

(-) The requirement for multiple independent models runs to identify privacy-safe samples and the need for sophisticated generative modeling adds computational overhead that may limit its applicability for resource-constrained settings.

**Questions:**

(1) The experiments focus solely on image datasets. How would the proposed approach perform with other data types, such as text, tabular data, or time-series data?

(2) The paper notes a decrease in test accuracy when privacy-risky fake data is introduced. Could the authors clarify the trade-off between privacy preservation and utility in more detail?

(3) The proposed method seems computationally expensive, involving multiple model runs and generative data transformations. Have the authors explored or considered techniques to optimize the computational requirements?

(4) The definitions of privacy-safe, privacy-risky, and privacy-vague samples are central to the paper. Could the authors provide a more formalized method for determining these privacy levels? Or do these definitions rely on empirical trials only?

---

### Note · Authors · 2024-11-12

I have read and agree with the venue's withdrawal policy on behalf of myself and my co-authors.